# The Flexible, Extended Coil of the PDZ-Binding Motif of the Three Deadly Human Coronavirus E Proteins Plays a Role in Pathogenicity

**DOI:** 10.3390/v14081707

**Published:** 2022-08-02

**Authors:** Dewald Schoeman, Ruben Cloete, Burtram C. Fielding

**Affiliations:** 1Molecular Biology and Virology Research Laboratory, Department of Medical Biosciences, University of the Western Cape, Private Bag X17, Bellville, Cape Town 7535, South Africa; dschoeman@uwc.ac.za; 2South African Medical Research Council Bioinformatics Unit, South African National Bioinformatics Institute, University of the Western Cape, Private Bag X17, Bellville, Cape Town 7535, South Africa; ruben@sanbi.ac.za

**Keywords:** human coronaviruses, envelope protein, PDZ-binding motif (PBM), homology-based modelling, docking, HADDOCK, protein-protein interaction, PALS1, pathogenesis, SARS-CoV-2

## Abstract

The less virulent human (h) coronaviruses (CoVs) 229E, NL63, OC43, and HKU1 cause mild, self-limiting respiratory tract infections, while the more virulent SARS-CoV-1, MERS-CoV, and SARS-CoV-2 have caused severe outbreaks. The CoV envelope (E) protein, an important contributor to the pathogenesis of severe hCoV infections, may provide insight into this disparate severity of the disease. We, therefore, generated full-length E protein models for SARS-CoV-1 and -2, MERS-CoV, HCoV-229E, and HCoV-NL63 and docked C-terminal peptides of each model to the PDZ domain of the human PALS1 protein. The PDZ-binding motif (PBM) of the SARS-CoV-1 and -2 and MERS-CoV models adopted a more flexible, extended coil, while the HCoV-229E and HCoV-NL63 models adopted a less flexible alpha helix. All the E peptides docked to PALS1 occupied the same binding site and the more virulent hCoV E peptides generally interacted more stably with PALS1 than the less virulent ones. We hypothesize that the increased flexibility of the PBM in the more virulent hCoVs facilitates more stable binding to various host proteins, thereby contributing to more severe disease. This is the first paper to model full-length 3D structures for both the more virulent and less virulent hCoV E proteins, providing novel insights for possible drug and/or vaccine development.

## 1. Introduction

Of the seven human (h) coronaviruses (CoVs) identified, the recent three, severe acute respiratory syndrome (SARS)-CoV-1, Middle East respiratory syndrome (MERS)-CoV, and SARS-CoV-2, are the most virulent and have caused severe outbreaks in the last two decades [1,2]. The remaining four hCoVs, HCoV-229E, -NL63, -OC43, and -HKU1, are less virulent and circulate continuously within the human population, peaking seasonally in different countries all over the world [3,4,5,6,7,8,9,10]. While these four generally cause less severe acute respiratory tract infections, such as the common cold, in immunocompetent persons, they can lead to more severe outcomes in immunocompromised persons, the elderly, and those with chronic, underlying medical conditions [6,11,12] The disease severity and immunopathology of SARS-CoV-1 and -2 infections have largely been associated with significant increases in the production of inflammatory cytokines, such as interleukin 1-beta (IL-1β), IL-6, and tumor necrosis factor alpha (TNF-α), which has been linked to pathological features such as pulmonary oedema, occasional lung epithelial damage, and diffuse alveolar damage (DAD) and can culminate in the lethal acute respiratory distress syndrome (ARDS) [13,14,15,16,17,18]. It is unclear, though, why HCoV-229E, -NL63, -OC43, and -HKU1 generally cause less severe disease in immunocompetent persons than SARS-CoV-1, -2, and MERS-CoV.

Clinical and pathological features such as DAD, accompanied occasionally by extensive damage to lung epithelium, and the dissemination of the virus to extrapulmonary tissues reported in SARS and coronavirus disease 2019 (COVID-19) patients gave rise to the possibility that SARS-CoV-1 and -2 viral proteins could potentially disrupt pulmonary epithelial architecture [19,20,21,22,23,24]. Over the last two decades it has become increasingly apparent that these pathological features, often seen in severe SARS-CoV-1 and -2 infections, can largely be attributed to two properties of the hCoV envelope (E) protein: its ion-channel (IC) activity and its ability to interact with host cell proteins through its postsynaptic density protein 95 (PSD95)/Drosophila disc large tumor suppressor (Dlg1)/zonula occludens-1 protein (ZO-1) (PDZ)-binding motif (PBM) [15,17,18,24,25,26,27]. The PBM of the prototypic SARS-CoV-1 E protein can bind to the PDZ domain of several host cell proteins [14,28,29,30], including protein-associated with Lin-7 1 (PALS1)—a protein normally associated with complexes found at intercellular tight junctions (TJs). This PALS1-E protein interaction redistributes PALS1 to the endoplasmic reticulum (ER)-Golgi intermediate compartment (ERGIC) where the E protein accumulates, delaying the formation of TJs and altering the formation of a uniform polarized monolayer of cells [24]. This led to a model in which the PALS1-E protein interaction was proposed to contribute to the desquamation of the alveolar wall observed in post mortem SARS-CoV-1-infected lung tissue and the extrapulmonary dissemination of the virus. 

Intercellular TJs and adherence junctions (AJs) are essential to establishing and maintaining epithelial cell polarity and, by extension, tissue architecture [31], which can provide physical protection against invading pathogens such as respiratory viruses [32,33]. Cell polarity is generally maintained by two cell complexes: the Par complex and the Crumbs (Crb) complex, the latter consisting of mammalian Lin-7 isoforms (MALS) 1, 2, and 3, PALS1, and PALS1-associated TJ protein (PATJ) [34,35]. Recent studies using C-terminal peptides of the SARS-CoV-2 E protein showed that it can interact with PALS1 similar to how the SARS-CoV-1 E protein does and reported that SARS-CoV-2 E had a higher affinity for PALS1 than SARS-CoV-1 E [36,37,38,39,40]. Furthermore, a recent study experimentally determined the hydrophobic pocket on the PALS1 protein to which the SARS-CoV-2 E peptide binds [39], while another experimentally demonstrated the residues involved in the interaction between PALS1 and peptides of the respective SARS-CoV-1, -2, and MERS-CoV E proteins, also specifying the types of bonds formed between the different interacting residues [40]. 

Over the past few decades, significant progress in structural biology and proteomics alongside computational advancements have resulted in more accurate predictions of protein structures and how protein complexes interact [41,42,43]. However, while much progress has been made, factors such as maintaining the native homeostatic environment of the studied protein, limitations to current laboratory techniques and methodologies, and the purification of protein complexes remain a challenge to generating high-quality, accurate models [44]. The complexity and largely hydrophobic nature of membrane proteins such as the CoV E protein also presents a challenge in studying their structure and dynamics experimentally [45,46,47]. In cases such as the hCoV E protein, where experimental data are sparse, template-based molecular modelling and molecular docking are computational techniques that can be employed to generate reasonably accurate structural models and predict interactions between proteins, which can then be explored experimentally [48]. 

The three-dimensional (3D) structural models generated in this study will provide insight into the pathogenicity of the more virulent hCoVs by predicting the secondary and tertiary structural fold of the E protein’s PBM, thereby establishing a structure–function relationship between the pathogenicity of the more virulent hCoVs vs. the less virulent ones. Molecular docking of the 3D models will demonstrate whether the less virulent hCoV E proteins can interact with PALS1 in a way similar to that of the more virulent ones and could shed light on both the disparate virulence of hCoVs and whether the potential PBMs of less virulent hCoVs are actually functional (Figure 1). Finally, the molecular dynamics (MD) simulations of the E protein models may provide insight into whether all hCoV E proteins share a uniform topology when embedded in lipid membranes, while also comparing the dynamic behavior of the E protein between the more virulent and less virulent hCoVs.

## 2. Materials and Methods

### 2.1. Molecular Modelling

#### Template Selection, Model Construction, and Quality Assessment

Full-length, 3D protein models were constructed for the more virulent SARS-CoV-1, -2, and MERS-CoV and the less virulent HCoV-229E and HCoV-NL63 E proteins using MODELLER software [49,50]. This was required to perform protein–protein docking studies to better understand the basis of the hCoV E protein PBM and PALS1 interaction and determine a possible disparity between the virulence of these hCoVs.

Two partial nuclear magnetic resonance (NMR)-resolved structures for the SARS-CoV-1 E protein (PDB IDs: 5X29 and 2MM4) were obtained from the Research Collaboratory for Structural Bioinformatics (RCSB) protein data bank (PDB) [51,52]. While both structures lacked some of the C-terminus, including the PBM, they spanned residues 8–65 and covered 76% of the total residues. Template 5X29 was used to generate full-length 3D models of the E proteins for SARS-CoV-1, -2, and HCoV-229E, whereas template 2MM4 was used construct the MERS-CoV model as it showed higher sequence identity to MERS-CoV compared with template 5X29. The predicted structure of HCoV-229E was used as the template for modelling the HCoV-NL63 E protein as it shared a higher amino acid sequence identity with HCoV-229E. The routinely used python script align2d.py in MODELLER was used to perform an alignment prediction between each of the hCoV E protein sequences and their respective template sequences [49,50]. Thereafter, a full-length 3D model was built for each hCoV E protein using the model-ligand.py script, and MODELLER [49,50] automatically predicted the structure of the missing C-terminal residues, which were compared with the C-terminal of experimentally resolved E peptides. 

The phi (φ) and psi (ψ) dihedral angle parameters for the Ramachandran plot was calculated for each of the predicted protein models using the PROCHECK webserver [53]. The root mean square deviation (RMSD) was calculated by performing a structural alignment between the predicted model structure and the homologous template protein to assess if any structural deviation exists within the main chain atoms of the two protein structures. Additional validation analysis was performed to determine the overall quality of the predicted models. The in-built assessment tools of MODELLER generated GA341 scores [54,55], which is a composite score that assesses the fold of the structure. The ProSA (protein structure analysis) webserver [56,57] was also used to calculate Z-scores, which measure the divergence of total energy of the structure with respect to an energy distribution derived from random conformations. The predicted 3D structures were visualized using PyMol molecular graphics software (version 2.5.2) [58].

### 2.2. High Ambiguity Driven DOCKing (HADDOCK): Molecular Docking 

The experimentally resolved (X-ray) protein complex (PDB ID: 7NTK) was obtained from the RCSB PDB and consists of the last eight residues (8-mer) of the SARS-CoV-2 E peptide (RVPDLLV) in complex with a partial PALS1 protein [40]. While the PALS1 protein contains three domains, i.e., the PDZ, sarcoma homology 3 (SH3), and guanylate kinase (GK) domains, the PALS1 protein in the 7NTK complex only contains the PDZ domain (residues 255–336), which has been shown to be sufficient for interaction with the hCoV E protein [24,40]. The pdb file for the 7NTK complex was separated into the partial PALS1 protein and the SARS-CoV-2 E 8-mer experimental peptide. 

The High Ambiguity Driven DOCKing (HADDOCK) 2.4 webserver (https://wenmr.science.uu.nl/haddock2.4/; accessed on 12 June 2022) [59] was used to dock the hCoV E protein models to the respective PALS1 protein. The experimental 8-mer SARS-CoV-2 E peptide (from 7NTK) was docked to the partial PALS1 protein structure to validate our docking parameters: to determine if HADDOCK could reproduce the experimental binding pose of the 7NTK complex. We also docked 8-mer and 18-mer peptides from the other E protein models—8-mer C-terminal peptides were selected, similar to Javorsky, et al. [40], while 18-residue C-terminal peptides (18-mer) were used to determine if a longer peptide might affect its interaction with PALS1. Briefly, the partial PALS1 structure and respective E peptides were uploaded to the HADDOCK webserver as input data, selecting “peptide” or “protein” as the “kind of molecule” when docking the 8-mer or 18-mer E peptides, respectively. Residues L267, G268, A269, V271, and R272 (PALS1) and the last four residues of the respective E peptides (PBM) were selected as input parameters for HADDOCK to specify the active residues to be used in the docking, thereby simulating the same residues experimentally determined by Javorsky, et al. [40]. All other docking parameters were kept at default, and transferable interaction potential with 3 points (TIP3P) water molecules was selected as the solvent to refine all the models. 

### 2.3. Interaction Analysis: PyMol and Protein-Ligand Interaction Profiler (PLiP)

Docked protein clusters generated by HADDOCK were viewed in PyMol and selected for further analysis based on how closely the cluster resembled the conformation of the SARS-CoV-2 E 8-mer peptide in the 7NTK experimentally resolved protein complex and if the docked peptides occupied the same binding site on PALS1. Those clusters were submitted to the protein-ligand interaction profiler (PLiP) webserver (https://plip-tool.biotec.tu-dresden.de/plip-web/plip/index; accessed on 12 June 2022) [60] and further selected based on those that reflect both the most similar interacting residues and the types of interactions (hydrophobic, hydrogen bonds, or ionic interactions) that occur between the 7NTK PALS1 protein and SARS-CoV-2 E 8-mer peptide as reported by Javorsky, et al. [40]: hydrogen bonds between L267, G268, A269 (PALS1), and V75 (SARS-CoV-2 E PBM); two hydrogen bonds between V271 (PALS1) and L73 (SARS-CoV-2 E PBM); and an ionic interaction between R272 (PALS1) and D72 (SARS-CoV-2 E PBM). If more than one docked cluster fulfilled these criteria, the E peptide of each cluster was superimposed on the SARS-CoV-2 E 8-mer experimental peptide, and the cluster with the lowest RMSD value was selected. The RMSD is the most commonly accepted measure used to demonstrate the similarity between the backbone atoms of two superimposed structures [61,62], with an RMSD of less than 2 Å indicating that all backbone elements are positioned correctly [63,64]. These criteria formed the basis of our system for determining the docked protein cluster with a modelled E peptide that adopts a similar conformation and that occupies the same binding site on PALS1, thereby most closely resembling the 7NTK protein complex (Figure 2).

### 2.4. Molecular Dynamic (MD) Simulations of the hCoV E Proteins in POPC Lipid Bilayer

Five simulation systems comprising the SARS-CoV-1, -2, MERS-CoV, HCoV-229E, and HCoV-NL63 E proteins were constructed using the Chemistry at Harvard Macromolecular Mechanics (CHARMM) graphical user web interface (GUI; CHARMM-GUI) solution builder (https://www.charmm-gui.org/; accessed on 12 June 2022). All five simulations were performed using the Groningen Machine for Chemical Simulations (GROMACS)-2019 package [65] along with the modified CHARMM36 (CHARMM36m) all-atom force field [66]. All the systems were embedded in a 1-palmitoyl-2-oleoylphosphatidylcholine (POPC) lipid bilayer [36] consisting of 120 lipid molecules (60 in the upper and 60 in the lower leaflet). The POPC lipid bilayer was chosen because it is the most abundant lipid in mammalian and eukaryotic membranes [67,68], and because the E protein is a membrane protein [48,69], this system would allow us to obtain a more realistic simulation of the protein. Each system was solvated with TIP3P water molecules in a cubic box of at least 10 Å of water between the protein and edges of the box at a concentration of 0.15 M KCl. To neutralize the positive and negative charges of the systems for SARS-CoV-1, -2, MERS-CoV, HCoV-229E, and HCoV-NL63, 22, 16, 29, 20, and 23 potassium (K^+^) ions and 22, 18, 30, 23, and 18 chloride (Cl^−^) ions were added to each system, respectively, using the Monte Carlo method [70]. Each of the systems underwent 50,000 steps of the steepest descents energy minimization to remove close van der Waals force contacts. All five systems were subsequently subjected to a two-step equilibration phase, namely, constant number of particles, volume, and temperature (NVT) for 500 ps to stabilize the temperature of the system and a short position restraint constant number of particles, pressure, and temperature (NPT) for 500 ps to stabilize the pressure of the system by relaxing the system and keeping the protein restrained. For the NVT simulation, the system was gradually heated by switching on the water bath and V-rescale temperature coupling [71,72] was used, with constant coupling of 0.1 ps at 300 K under a random sampling seed. For NPT, the Parrinello–Rahman pressure coupling [73] was turned on with constant coupling of 0.1 ps at 300 K under conditions of position restraints (all-bonds). For both NVT and NPT, electrostatic forces were calculated using the particle mesh Ewald method [74]. All five systems were subjected to a full 100 ns simulation without restraints. 

### 2.5. Trajectory Analysis, Principal Component Analysis (PCA), and Lipid Bilayer Analysis

The trajectory files were analyzed using GROMACS utilities. The RMSD was calculated using gmx rmsd for the protein backbone atoms, and root mean square fluctuation (RMSF) for the protein atoms was calculated using gmx rms. The radius of gyration (Rg) for the backbone atoms was calculated using the gmx gyrate tool, while the solvent accessible surface area (SASA) for the protein atoms was calculated using gmx sasa. Principal component analysis (PCA) was used as a statistical method to reduce the complexity of the data set to identify the most relevant movements of the protein. Briefly, for PCA, we first calculated the covariance matrix by removing translated and rotational movement using gmx covar and then extracted the eigenvectors for the first two principal components that contributed to the largest movement of the protein using gmx anaeig. We characterized the simulated POPC lipid bilayer by calculating several parameters. This lipid bilayer analysis comprised of different calculations with the corresponding GROMACS utility indicated in paratheses: the area per lipid (gmx energy) to analyze the X and Y box dimensions, the bilayer thickness (gmx density) for the phosphate atom headgroups, the lateral diffusion coefficients (gmx mean square displacement (msd)) for the lipid head groups, and the deuterium order parameters (gmx order) for all carbons in the acyl chains. The average number of hydrogen bonds were also calculated between the E protein and POPC lipids (gmx hbond), and to determine interactions between the PBM and POPC lipids we extracted the final structure frame at 100 ns (gmx trjconv). The “find: polar contacts” option in PyMol was used to determine these polar interactions.

## 3. Results

### 3.1. Homology Modelling and Quality Assessment

#### 3.1.1. More Virulent hCoVs: SARS-CoV-1, -2, and MERS-CoV

Both the SARS-CoV-1 and -2 E proteins shared a 91% amino acid sequence identity with template 5X29 (Appendix A), and their 3D models showed three α-helices and four coil regions, with the C-terminal PBM adopting a coil region (Figure 3A,B). In comparison, the MERS-CoV E protein only shared a 35% sequence identity to template 2MM4 (Appendix A), and while the 3D model showed only two α-helices and three coil regions, the PBM also adopted a coil region (Figure 3C). We superimposed the C-terminals of our SARS-CoV-1, -2, and MERS-CoV E protein models onto the C-termini of experimentally resolved SARS-CoV-1 and -2 E proteins (PDB IDs: 7NTJ, 7NTK, and 7M4R) to determine whether the PBM of our models adopted the appropriate conformation. We found very little structural difference between the modelled PBMs of our SARS-CoV-1, -2, and MERS-CoV E protein models and that of the experimentally resolved E proteins (Appendix A). This showed that the PBM of our SARS-CoV-1, -2, and MERS-CoV E protein models adopted the appropriate conformation. Quality assessment of the three models revealed a somewhat similar percentage of residues in the most favored regions and the disallowed regions of the Ramachandran plot. For the SARS-CoV-1, -2, and MERS-CoV E models, respectively, 90%, 88.6%, and 95.8% of the residues were in the most favored regions. The SARS-CoV-2 model had 0% of its residues in the disallowed regions, whereas both the SARS-CoV-1 and MERS-CoV models had 1.4% of their residues in the disallowed regions. Compared with the 5X29 template, the RMSD analysis of the SARS-CoV-1 and -2 E protein models showed 0.698 Å and 2.155 Å differences, respectively, whereas the MERS-CoV E protein model showed a difference of 1.458 Å against the 2MM4 template. The ProSA Z-scores of the different E protein models were greater than 0, with SARS-CoV-1 having the highest (0.65) followed by SARS-CoV-2 (0.59), and MERS-CoV (0.25) had the lowest. The GA341 score was highest for the SARS-CoV-1 model (1), followed by SARS-CoV-2 (0.95), with the MERS-CoV model having the lowest score (0.41).

#### 3.1.2. Less Virulent hCoVs: HCoV-229E and HCoV-NL63

The HCoV-229E E protein shared a 29% sequence identity with the amino acid sequence of template 5X29 (Appendix A), whereas the HCoV-NL63 E protein shared a higher sequence identity with the homologous HCoV-229 E template (Appendix A), which is why it was selected as the template for constructing the HCoV-NL63 E model. Both 3D models showed four α-helices and four coil regions, with the C-terminal PBM adopting one of the α-helical turns (Figure 3D,E), and V74 from both HCoV-229E and HCoV-NL63 shared conservation with V78 from template 5X29. The quality assessment of both models showed very similar percentages of residues in the most favored regions and the disallowed regions of the Ramachandran plot. For HCoV-229E and HCoV-NL63 E, respectively, 91.5% and 91% of the residues were in most favored regions, while both models had 0% of residues in the disallowed regions. The RMSD analysis showed a 1.776 Å difference between the HCoV-229E E protein model and template structure 5X29, whereas the difference between the HCoV-NL63 E protein model and the HCoV-229E template was 1.217 Å. The HCoV-229E E protein model had the highest ProSA Z-score of 0.81, whereas the E protein model of HCoV-NL63 was the only model with a negative score of −0.46. The GA341 score of the HCoV-NL63 E protein model (0.33) was higher than that of the HCoV-229E model (0.15).

### 3.2. Molecular Docking and Interaction Analysis

Docking and interaction analysis of the SARS-CoV-2 8-mer experimental E peptide bound to the PALS1 PDZ domain from the 7NTK complex confirmed the interactions reported by Javorsky, et al. [40], i.e., hydrogen bonds formed between the terminal PBM residue V75 and PALS1 residues L267, G268, and A269; between PBM L73 and PALS1 V271; and between PBM D72 and PALS1 residues V271 and S281 (Figure 4A and Table 1). We also found several hydrophobic interactions between PBM residues L73, L74, and V75 and PALS1 residues T270, V271, V284, V314, F318, and L321, along with a single ionic interaction between PBM residue D72 and PALS1 R272. The docked 8-mer and 18-mer E peptides produced from the 3D E protein models were all compared with this interaction analysis of the 7NTK SARS-CoV-2 E-PALS1 complex.

Docking of the modelled SARS-CoV-1 E 8-mer peptide largely showed similar interactions with the corresponding PALS1 residues with some minor differences in the positions of the interacting residues (Figure 4B and Table 1). Many of the hydrogen bonds and hydrophobic interactions that formed between the PBM and PALS1 residues largely corresponded to the ones formed in the experimental 7NTK protein complex. A single ionic interaction was retained between PBM residue D73 and PALS1 R272. The modelled SARS-CoV-1 18-mer E peptide, however, adopted an orientation whereby residues upstream of the PBM also occupied the PALS1 binding site (Figure 4C and Table 2). Regardless, most of the residue positions where hydrogen bonds and hydrophobic interactions were formed were retained, and a single ionic interaction formed between the PBM residue D73 and PALS1 residue R282.

The docking of both the 8-mer and 18-mer modelled SARS-CoV-2 E peptides resembled the interactions of the 7NTK complex the most, sharing many of the same interacting residues at the same positions (Figure 4D,E and Table 1 and Table 2). A notable difference is that PBM residue D72 of the 8-mer peptide formed an ionic interaction with PALS1 residue R282, whereas the 18-mer peptide retained the ionic interaction with PALS1 residue R272 and formed an additional one with E274.

The PBM of the modelled MERS-CoV 8-mer E peptide, while different in composition, exhibited very similar binding tendencies to the docked modelled SARS-CoV-1 and -2 E peptides (Figure 4F and Table 1). It occupied the same PALS1 binding site, and interactions were still largely confined to the PBM. Many of the hydrogen bonds and hydrophobic interactions formed between the PBM and PALS1 residues were similar to those of the 7NTK complex, and a single ionic interaction between PBM residue D79 and PALS1 residue R272 was retained. The modelled 18-mer MERS-CoV E peptide largely retained the same hydrogen bonds as the modelled 8-mer peptide, but only two hydrophobic interactions were formed between the peptide and PALS1 (Figure 4G and Table 2). No ionic interaction was detected, but a π-stacking interaction was found between PBM residue W81 and residue F318 of PALS1.

The HCoV-229E 8-mer E peptide occupied the same binding site as the SARS-CoV-2 8-mer experimental E peptide in the 7NTK complex, with interactions still predominantly confined to PBM residues (Figure 5A and Table 1). Hydrogen bonds were found only between the terminal PBM residue F77 and PALS1 residues L267, G268, and A269, with a few hydrophobic interactions between F77 and other PALS1 binding site residues. While no ionic interaction was found, the residue upstream of the PBM residue F70 did form a π-cation interaction with PALS1 R272. The PBM residue F77 on the modelled HCoV-229E 18-mer E peptide formed hydrogen bonds with PALS1 residues L267, G268, and A269, and residues upstream of the PBM formed additional hydrogen bonds with other PALS1 residues (Figure 5B and Table 2). More hydrophobic interactions were found between the 18-mer peptide and PALS1 binding site residues, most of which were still attributed to PBM residue F77. No ionic interaction was detected, but a π-stacking interaction was found between the E peptide residue Y61 and PALS1 F318.

The PBM of the modelled HCoV-NL63 8-mer E peptide occupied the same binding site, with most of the hydrogen bonds formed attributed to the terminal E PBM residue V77 (Figure 5C and Table 1). Some hydrophobic interactions were found between PBM residues V74, L75, and V77 and PALS1 binding site residues, but no ionic or π-cation interactions were found. Hydrogen bonds in the modelled 18-mer HCoV-NL63 E peptide predominantly formed between PBM residue N76 and PALS1 residues L267, G268, and A269, while hydrophobic interactions centered around E peptide residues E73, V74, and V77 and PALS1 residues F318 and L321 (Figure 5D and Table 2). No ionic interaction was found, but a π-stacking interaction was found between the E peptide residue Y61 and PALS1 residue F318.

### 3.3. Molecular Dynamics Simulation Analysis

The RMSD analysis of the five hCoVs systems indicated that all five systems reached equilibrium after 60 ns (Figure 6A). The mean and standard deviation (SD) for the backbone RMSD were the largest for HCoV-229E followed by SARS-CoV-1, MERS-CoV, HCoV-NL63, and SARS-CoV-2 at 0.94 ± 0.21 nm, 0.71 ± 0.12 nm, 0.65 ± 0.12 nm, 0.56 ± 0.11 nm, and 0.40 ± 0.05 nm, respectively (Figure 6A). For the RMSF analysis, the largest flexibility was observed for HCoV-229E, followed by SARS-CoV-1, HCoV-NL63, and MERS-CoV, while the lowest flexibility was observed for SARS-CoV-2 (Figure 6B). Additionally, the mean and SD for the protein residues were the lowest for SARS-CoV-2 (0.15 ± 0.06 nm), followed by HCoV-NL63 (0.27 ± 0.13 nm), MERS-CoV (0.27 ± 0.33 nm), SARS-CoV-1 (0.44 ± 0.21 nm), and HCoV-229E (1.27 ± 1.48 nm). Plotting the 3D structure for the E protein of each hCoV using PyMol indicated that the PBM was flexible for SARS-CoV-1, -2, and MERS-CoV, while for HCoV-229E and HCoV-NL63, the PBM was stable (Figure 3A–E). Furthermore, the 3D conformations of the PBM of HCoV-NL63 and HCoV-229E adopted a single α-helical conformation in contrast with SARS-CoV-1, -2, and MERS-CoV, which had an extended coil at the C-terminal PBM (Figure 3A–E). The mean and SD for the Rg were the highest for the MERS-CoV and HCoV-229E systems, 2.07 ± 0.05 and 2.04 ± 0.10 nm, respectively, followed by HCoV-NL63 and SARS-CoV-1 at 1.72 ± 0.06 and 1.83 ± 0.05 nm, respectively; SARS-CoV-2 had the lowest mean and SD at 1.75 ± 0.03 nm (Figure 6C). Similarly, the mean and SD of the SASA was the highest for MERS-CoV and HCoV-229E, followed by SARS-CoV-1, HCoV-NL63, and SARS-CoV-2: 87.14 ± 2.32, 83.95 ± 2.34, 79.06 ± 1.74, 77.91 ± 1.87 nm^2^, and 75.42 ± 1.44 nm^2^, respectively (Figure 6D). The PCA indicated that SARS-CoV-2 had the smallest covariance matrix after diagonalization of 5.93 nm^2^ compared with HCoV-NL63, MERS-CoV, SARS-CoV-1, and HCoV-229E at 14.09, 25.49, 32.59, and 54.58 nm^2^, respectively (Appendix A).

The average area per lipid reached convergence and fluctuated around a stable value for each system (Figure 7 and Table 3). The means and SDs for each of the various lipid bilayer properties are summarized in Table 3. The bilayer thickness was found to be the lowest for the SARS-CoV-2 system (4.25 ± 2.48 nm) followed by SARS-CoV-1 (4.39 ± 2.87 nm), then HCoV-NL63 (4.46 ± 2.60 nm) and HCoV-229E (4.65 ± 2.71 nm), with the MERS-CoV system (5.78 ± 3.37 nm) being the highest. The lateral diffusion coefficient was the lowest for the HCoV-NL63 (0.34 ± 0.05 cm^2^/s) system, followed by SARS-CoV-1 (0.37 ± 0.02 cm^2^/s), then SARS-CoV-2 (0.43 ± 0.09 cm^2^/s) and HCoV-229E (0.46 ± 0.02 cm^2^/s), while the MERS-CoV system had the highest, reaching 0.84 ± 0.01 cm^2^/s. The deuterium order parameters for the POPC carbon acyl chains sn-1 and sn-2 indicated similar results (Appendix A). For all the systems of chain 1, atom 2 had the lowest deuterium order parameter value while atoms 3–7 had higher values, suggesting that atoms 3–7 are not undergoing any large structural changes. For chain 2, the terminal carbons (1 and 7) had lower deuterium order parameter values, while the central atoms 2–6 had higher values, suggesting that the central atoms are maintaining their structure. The average number of hydrogen bonds between the E protein systems and POPC lipids was the highest between SARS-CoV-2 and POPC (13.04) followed by HCoV-NL63 (12.64), HCoV-229E (11.89), SARS-CoV-1 (10.58), and MERS-CoV (9.22). Calculating the number of interactions between the PBM and POPC lipids indicated that SARS-CoV-2 formed two polar contacts (Appendix A) and HCoV-NL63 (Appendix A) formed one. No polar contacts were identified between POPC and SARS-CoV-1, MERS-CoV, and HCoV-229E.

## 4. Discussion

Membrane proteins such as hCoV E play an important part in viral assembly and the release of virions, aspects reliant on the protein structure and topology [75,76,77]. In addition to these functions, hCoV E is also a major contributor to coronaviral pathogenesis and clinical pathology owing to its IC activity and ability to interact with host cell proteins through its PBM [14,15,17,18,24,25,26,27,28,29]. Of all the hCoV E proteins, only partial structures for SARS-CoV-1 and -2 have been resolved experimentally and deposited in the RCSB PDB. The PDB IDs 5X29 and 2MM4 were used as templates in this study because they are the longest experimentally resolved structure (spanning residues 8–65) but are both from SARS-CoV-1 E [51,52]. The longest SARS-CoV-2 E experimentally resolved structure only spans the transmembrane domain (TMD), residues 8–38 [78], and lacked the C-terminal, which would not have allowed for making meaningful interpretations about the full-length protein. Therefore, 5X29 and 2MM4 were selected as templates to construct a 3D full-length model of the SARS-CoV-1 E protein, which was subsequently used to construct the 3D full-length models for the other hCoV E proteins. 

The 91% sequence identity between template 5X29 and the SARS-CoV-1 and 2 E proteins (Appendix A) is to be expected because of the very high amino acid sequence similarity between SARS-CoV-1 and -2 E [17,79] and because template 5X29 spans residues 8–65 of the SARS-CoV-1 E UniProt reference sequence (UniProt accession number: P59637). The 3D structures predicted for the SARS-CoV-1 and -2 E proteins successfully satisfied the Ramachandran plot dihedral angle distributions, and based on the RMSD between both the SARS-CoV-1 and -2 E proteins and template 5X29, the correct protein folds were assigned to the respective protein sequences. The 35% sequence identity between the MERS-CoV E protein and template 2MM4 suggests a medium amount of sequence homology (Appendix A) but is also to be expected given the much lower amino acid sequence similarity between SARS-CoV-1 E and MERS-CoV E [17,79]. Nevertheless, the RMSD calculated between the MERS-CoV E 3D structure and 2MM4 showed a low deviation, suggesting high structural similarity that still allows for reliable domain prediction. Despite the moderate sequence identity between the HCoV-229E E protein and the 5X29 template, the protein model from this pairwise sequence alignment will provide a medium accuracy model [80] that is still useful for domain comparisons. The RMSD between HCoV-229E E and the homologous template structure 5X29 is also low enough to suggest very little deviation between the two structures. The RMSD analysis between HCoV-NL63 E and the homologous template structure (HCoV-229E protein model) similarly indicated a difference that suggests very little deviation between the generated model and the homologous template structure. The ProSA Z-scores of all the models, except for HCoV-NL63, suggest a high confidence in the structures, but all structures were still in the range of structures with similar conformations. The GA341 score for all the 3D models was also all higher than 0, suggesting the correct fold was assigned to the models.

Over the course of the COVID-19 pandemic, the SARS-CoV-2 spike (S) protein has incurred many mutations [81,82,83,84,85], raising the concern that other viral proteins might also have incurred mutations. However, several studies have analyzed sequenced genomes obtained from COVID-19 databases and reported minimal mutations in the SARS-CoV-2 E gene [85,86,87,88,89,90,91], remarking that the E protein and its properties are highly conserved [90,91,92,93,94]. We therefore conclude that the full-length, 3D structures predicted for all five of the E proteins successfully passed quality assessment, and based on the low RSMD values, we are confident that the correct protein folds were assigned to the respective protein sequences. Additionally, despite the absence of experimentally resolved structures for the HCoV-229E and HCoV-NL63 E proteins, we are still confident in the accuracy of these two models. We also attempted to construct full-length 3D models of HCoV-OC43 and HCoV-HKU1 E but excluded them from this study since these two E proteins shared a much lower sequence identity with the other hCoV E proteins, which resulted in lower accuracy and less reliable models of HCoV-OC43 and HCoV-HKU1 E. 

Other studies have also modelled hCoV E proteins, but they have focused exclusively on the more virulent hCoV E proteins and excluded the less virulent ones [36,92,93,95]. Some studies have only constructed partial models of SARS-CoV-2 E using 5X29 as template [95] or SARS-CoV-1, -2, and MERS-CoV E proteins from the 2MM4 template [92]. One study, interestingly, used a full-length SARS-CoV-2 E model that was constructed from machine learning and physics-based refinement [93], but the model was not published. Only one other study constructed full-length models of SARS-CoV-1 and -2 E similar to our models but used only 2MM4 as template [36]. Thus, to the best of our knowledge, our study is the first to construct full-length, 3D models of the E protein for both the more virulent and the less virulent hCoVs. Furthermore, barring the missing residues from 5X29 and 2MM4, all five of our E protein models share exactly the same structural features as those found in these previous studies, i.e., the N-terminus forms a coiled structure and the TMD adopts an α-helical conformation, with a second α-helix (H2) after the TMD [39,96,97], demonstrating that the secondary structure arrangement of our models is comparable with predictions from previous studies. 

All our models adopted an α-helix in the region corresponding to the TMD, which inserts into the membrane of the ERGIC where the E protein accumulates and coronaviral assembly occurs [94,96,97,98,99]. Yet neither our MERS-CoV E model nor the partial 2MM4-based MERS-CoV E model generated by Aldaais, et al. [92] adopts the amphipathic H2 α-helix immediately adjacent to the TMD. This H2 α-helix is, nevertheless, also present in the SARS-CoV-1, -2, HCoV-229E, and HCoV-NL63 E proteins, suggesting that H2 might not play a role in the E protein-mediated pathogenesis but could be linked to other E protein functions such as viral assembly. While the scarcity of MERS-CoV data and lack of an experimentally resolved E protein structure make it difficult to confirm the significance of the missing helix at this position, in silico coarse-grain (CG) and atomistic simulations do suggest that the second and third α-helices (H2 and H3) of the SARS-CoV-2 E protein are important in inducing membrane curvature [93]. Moreover, since amphipathic helices (AHs) in both cellular and viral proteins have been shown to be important in membrane curvature [100,101,102,103], the lack of an H2 in the MERS-CoV E model suggests that MERS-CoV E could be structurally defective in its ability to induce membrane curvature, which would impede the assembly and release of MERS-CoV virions. Since MERS-CoV was much less effective at human–human transmission than the two other more virulent hCoVs [104,105,106,107], it is possible that this could be linked to its limited transmissibility. While the lack of this H2 would not render it incapable of infection and replication, it could impede the ability of MERS-CoV to assemble and release new virions effectively, possibly leading to fewer viruses shed by infected persons, thereby limiting its transmissibility to small, close-contact groups [106,108]. However, CoV nonstructural proteins (nsps), some of which may have overlapping functions, could also potentially compensate, at least partially, for the lack of an H2 α-helix in the MERS-CoV E protein [109]. Alternatively, MERS-CoV E could simply induce membrane curvature by a different mechanism. Without experimental evidence, this remains to be determined.

Comparing the structural aspects of the E protein models, the PBM of SARS-CoV-1, -2 and MERS-CoV E proteins adopted a more flexible extended coil region, while the PBM of HCoV-229E and HCoV-NL63 E is characterized by a less flexible α-helical turn, suggesting that the flexibility of the E protein PBM region might be a feature distinguishing the more virulent hCoVs from the less virulent ones. The RMSF average from the MD analysis showed agreement with the predicted 3D structures of the E proteins generated by MODELLER. The PBM of the SARS-CoV-1, -2 and MERS-CoV E proteins adopted an extended coil at the C-terminal, while the E proteins of HCoV-229E and HCoV-NL63 adopted a stable alpha helical conformation at their PBM. Furthermore, MD analysis of the entire protein dynamics showed that SARS-CoV-2 E was more stable than the E proteins of SARS-CoV-1, MERS-CoV, and HCoV-NL63, while the HCoV-229E E protein exhibited higher flexibility, suggesting that it is the least stable of the five hCoV E proteins. This is based on RMSD, RMSF, Rg, and SASA parameter calculations. The PCA confirmed the protein dynamics, showing that SARS-CoV-2 E exhibited the least randomized movements followed by HCoV-NL63, MERS-CoV, SARS-CoV-1, and HCoV-229E showing more concerted movements throughout the phase space. This would certainly be interesting to confirm and warrants exploring experimentally with in vitro and in vivo follow-up studies. Analysis of the POPC lipid bilayer indicated that the lipid structure remained stable throughout the simulation, while the SARS-CoV-2 E protein made more contacts with the POPC lipid membrane, explaining its higher stability within the lipid bilayer [110,111]. However, the PBM of SARS-CoV-2 E only made two polar contacts with POPC lipids in the final 200 ns frame, suggesting its high mobility.

Interestingly, De Maio, et al. [36] used the Homology Modeling protocol, Prime, of the Schrodinger suite [112] to generate a full-length, 3D model of the SARS-CoV-2 E protein. The PBM of their model, like that of our E protein models for the more virulent hCoVs, also adopted a flexible coil. Furthermore, when we compared the PBMs of our SARS-CoV-1, -2, and MERS-CoV E protein models with those of experimentally resolved SARS-CoV-1 and -2 E proteins, we found that our models adopted the same flexible coil conformation (Appendix A). This led us to conclude that the conformation of the E protein PBM could quite possibly be a structural feature that allows the more virulent hCoVs to interact with a wider range of host proteins, which could explain their increased pathogenicity and disease severity. By comparison, the less flexible α-helical turn of the less virulent hCoV E protein PBM could likely be an impeding characteristic, limiting or even possibly preventing interactions with host proteins, which might explain the more limited pathogenicity of these hCoVs. Several host cell proteins have so far been reported to interact with the E protein of the three more virulent hCoVs [14,24,28,29], and more may yet be identified, but no such studies have been undertaken for the less virulent hCoVs.

While our system is based in silico, the generated 3D models and docking were based on experimentally determined, published results [43,81,82], and we validated our system by first docking the experimental SARS-CoV-2 8-mer E peptide to the PALS1 PDZ domain to replicate the experimentally determined interaction between SARS-CoV-2 E and PALS1. Although we only used the experimentally determined residues by Javorsky, et al. [40] to perform docking, it is very interesting that our interaction analysis detected interactions with PALS1 residues P266, L267, V284, F318, and L321 to varying degrees in our docked models (Figure 4 and Figure 5) (Table 1 and Table 2). We find this interesting because these residues are reported to be involved in forming the hydrophobic pocket on PALS1 to which the SARS-CoV-2 E protein binds, and the positioning of F318 between the SARS-CoV-2 E proteins L73 and V75 is considered an important recognition feature [39]. Several of our docked peptides share this recognition feature in that PBM residues are positioned on either side of PALS1 residue F318, and one/two of the PBM residues form a hydrophobic interaction with F318 on PALS1 (Table 1 and Table 2). Furthermore, the SARS-CoV-2 E PBM residues V75, L74, and L73 are reportedly involved in hydrophobic interactions with PALS1, but the corresponding PALS1 residues were not specified [40]. Our interaction analysis showed that these PBM residues form several hydrophobic interactions with the PALS1 residues reported to form the hydrophobic pocket of the PALS1 PDZ domain [39] (Table 1 and Table 2). Still, all our docked E peptides occupied the same binding site on the PALS1 PDZ domain as the experimentally determined interaction between the PALS1 PDZ domain and the 8-mer SARS-CoV-2 E peptide (Figure 4 and Figure 5) (Table 1 and Table 2) [40], thereby validating the docking protocol for our system.

Another notable aspect is the ionic interaction formed between the aspartic acid residue at the start of the PBM of all three of the more virulent hCoV E peptides and the PALS1 residue R272/R282. Such an interaction is absent from the less virulent hCoV E peptides and is likely due to their PBM starting with the nonpolar residue valine, as opposed to the negatively charged aspartic acid on the E protein of more virulent hCoVs. Given that ionic interactions are generally considered stronger than both hydrogen bonds and hydrophobic interactions [113], the presence of the negatively charged aspartic acid at this position in the PBM might also be something that allows the more virulent hCoV E proteins to form a more stable interaction with PALS1 to facilitate their increased viral pathogenicity. In fact, the native binding partner of PALS1, Crumbs (Crb), also forms an ionic interaction with R282 on PALS1 through its negatively charged E1403 residue [35,40], similar to what we found in our docked SARS-CoV-2 E 8-mer peptide. The ERLI motif (**E**1403, **R**1404, **L**1405, **I**1406) of the host protein Crb is functionally similar to the E protein’s PBM and allows Crb to occupy the same binding site on the PALS1 PDZ domain [38,41,43]. Furthermore, the last residue of the Crb ERLI motif (I1406) forms hydrogen bonds with PALS1 residues L267, G268, and A269, and its E1403 interacts with PALS1 residues T270 and S281 through hydrogen bonds [35]. Similarly, the last residue of each of our modelled E peptide PBMs also formed hydrogen bonds with PALS1 residues L267, G268, and A269. Moreover, the aspartic acid of the more virulent SARS-CoV-2 (D72) and MERS-CoV (D79) E peptides also shows hydrogen bonding to the PALS1 S281, whereas the SARS-CoV-1 (D76) E peptide does not, while the SARS-CoV-2 E peptide (D72) is the only one to form a hydrogen bond with PALS1 T270. Zhu, et al. [114] recently also resolved a SARS-CoV-2 E peptide bound to the PDZ domain of PALS1 (residues 238–336; PDB ID: 7QCS), and our interaction analysis also largely correspond to their experimental findings. Using the number of bonds formed between the E peptide and PALS1 PDZ domain as a measure of stability, this could potentially explain why SARS-CoV-1 was less severe than MERS and COVID-19 and why COVID-19 appears to be more severe than both MERS and SARS diseases were. Interestingly, Lo Cascio, et al. [38] reported that the SARS-CoV-1 E peptide (containing the PBM) did not remain attached to PALS1 for longer than 150–200 ns, even after four MD simulation runs. The SARS-CoV-2 E peptide (also containing the PBM) did remain attached to PALS1, suggesting that while SARS-CoV-1 E does seem to interact with PALS1 in the same way that SARS-CoV-2 E does, the two E peptides do not interact with PALS1 for the same amount of time. This longer interaction between SARS-CoV-2 E and PALS1 could explain why SARS-CoV-1 and -2 exhibit similar pathologies but SARS-CoV-2 infections are more severe. Overall, the similarity between previous studies and the types of interactions between the specific E protein and PALS1 residues further validates the accuracy of the docked E peptides of our system. 

There is a rather clear difference in the number of bonds formed between the E peptides of the more virulent hCoVs and PALS1 and the number of bonds formed between PALS1 and the E peptides of the less virulent hCoVs. Generally, more bonds are formed between PALS1 and the E peptides of the more virulent hCoVs, possibly producing a more stable interaction, especially since many of the interactions are hydrogen bonds in addition to several hydrophobic interactions. In comparison, the E peptides of the less virulent HCoV-229E and HCoV-NL63 formed fewer hydrogen bonds and fewer hydrophobic interactions, suggesting that while they might be capable of binding to PALS1, the interaction might be less stable, or merely transient, than in the case of the more virulent hCoV E peptides. Nevertheless, our system still suggests that the SARS-CoV-1, -2, MERS-CoV, HCoV-229E, and HCoV-NL63 E proteins can occupy the same binding site on the PALS1 PDZ domain, albeit to different extents. This could explain why the less virulent hCoVs tend to be less likely to spread to extra-pulmonary sites, thereby being less likely to cause severe disease. If the HCoV-229E and HCoV-NL63 E proteins cannot bind to PALS1 as effectively and stably as those of SARS-CoV-1, -2, and MERS-CoV can, they might not be able to displace PALS1 from the Crb-tight-junction-complex and would consequently not be able to compromise the pulmonary epithelial barrier to the extent that the more virulent hCoV E proteins can.

Two recent studies used a different technique to measure the binding affinity between E peptides of the more virulent hCoVs and the PALS1 PDZ domain and reported conflicting results. One study used C-terminal E peptides of 12 amino acids long (12-mer) and found that SARS-CoV-1 E did not bind to the PALS1 PDZ domain, while SARS-CoV-2 E only bound weakly to PALS1 (K_d_ = 102 µM) [30]; the other study used 8-mer C-terminal E peptides and found that both SARS-CoV-1 and -2 E bound to the PALS1 PDZ domain with dissociation constants of 29.6 ± 2.3 µM and 22.8 ± 1.2 µM, respectively [40]. Both studies, however, reported that the MERS-CoV E peptide did not bind to the PALS1 PDZ domain [30,40]. Javorsky, et al. [40] proposed that the W81 of the MERS-CoV E PBM is too large and bulky to accommodate binding to the PDZ binding groove, whereas SARS-CoV-1 and -2 E, and Crb feature the smaller leucine residue that could facilitate binding to PALS1. Conversely, Caillet-Saguy, et al. [30] suggested that the differences in binding of SARS-CoV-1 and 2 and MERS-CoV E proteins to PALS1 and other host proteins are due to their differences in PBM classes. Class II PBMs, as seen in the SARS-CoV-1 and -2 E proteins, are characterized by the motif: Φ-X-Φ, where X is any amino acid and Φ is a hydrophobic residue, whereas class III PBMs, as seen in MERS-CoV E protein, have the motif: D/E-X-Φ. The authors attributed the hydrophobic character of class II PBMs to having sharp binding profiles but lower affinities for their target host proteins, while the presence of the tryptophan residue (W81) in the class III PBM of MERS-CoV E likely strongly contributed to the binding affinity of more of its host target proteins but at the expense of specificity. While the PBMs of the E proteins for the less virulent hCoVs have not specifically been classified or characterized, the HCoV-229E and HCoV-NL63 E proteins appear to adopt a class II PBM, similar to SARS-CoV-1 and -2, despite demonstrating a lower viral pathogenicity. Although the E proteins of HCoV-229E and HCoV-NL63 potentially also have a class II PBM, their lower viral pathogenicity could also, at least in part, be due to the lower number of hydrogen bonds and hydrophobic interactions and the absence of ionic interactions between their E proteins and PALS1. Indeed, both hydrogen bonds and hydrophobic interactions have been shown to be important in protein stability and at the interface of protein–protein interactions (PPIs) [115,116]. This combination of a less flexible alpha helix in their E protein PBM could possibly reduce the number of hydrogen bonds and hydrophobic interactions, while the lack of a negatively charged residue, such as aspartic acid, might hinder the formation of an ionic interaction with the PALS1 protein, ultimately impeding the ability of these hCoV E proteins from binding effectively to PALS1 [116]. Stodola, et al. [117] found that a mutated PBM of the HCoV-OC43 E protein impaired virion production in an epithelial cell line and attenuated the virus in neuronal cells. The authors, unfortunately, did not investigate specific viral-host PPIs, but their study does indicate that the less virulent hCoV E proteins appear to be capable of interacting with host cell proteins. While perhaps still capable of interaction with host proteins, our results suggest that the E proteins of HCoV-229E and HC-oV-NL63 are unable to bind to PALS1 as effectively as the E proteins of the more virulent hCoVs. However, this should be validated experimentally.

There are studies that have investigated the interaction between the hCoV E protein and PALS1 [24,42,43], syntenin [14], the tight junction protein ZO-1 [118], and several other host cell proteins [30]. However, these studies generally investigated only one of the more virulent hCoV E proteins at a time with the focus predominantly on the SARS-CoV-1 and -2 E proteins and their interactions with a particular host cell protein. They did not include the E proteins from any of the less virulent hCoVs. Conversely, our study generated full-length 3D models for all three of the more virulent hCoV E proteins as well as two of the less virulent hCoV E proteins, while the majority of papers that have also produced 3D models of the hCoV E protein have only generated partial models for SARS-CoV-1 and/or -2 [92,93,94,119]. We found only one paper that produced an E protein model for MERS-CoV, but it is still only a partial model [92]. We have not found any papers that produced a 3D model, partial or full-length, for any of the less virulent hCoV E proteins. While our study certainly would have been more comprehensive if we included the host proteins syntenin and ZO-1 in our docking and interaction analysis, we considered how well-characterized the viral-host PPI was, the number of PDZ domains the target host protein has, and the quality or availability of the host protein’s experimentally resolved structure in the RCSB PDB. The PPIs of syntenin and ZO-1 are not as well-characterized as the PALS1-E interaction, and syntenin has two PDZ domains and ZO-1 has three PDZ, whereas PALS1 only has one PDZ domain. Additionally, while studies that use in vitro [14,118] and in vivo systems [14] have the benefit of assessing these viral-host PPIs in more biologically relevant contexts, our in silico system allows us to first make predictions [120] and obtain better insight by simulating these virus-host PPIs, which can then be validated experimentally with in vitro and in vivo studies. Thus, our 3D models have the potential to be used for screening potential inhibitors of the E protein’s IC function [121,122,123]. They can also be used to predict the potential effects of different mutations on the pathogenicity of the virus. By simulating the mutation of a key residue, our models can predict how such mutations might affect the structure and/or function of the E protein and consequently the pathogenicity of the virus.

There are limitations to our study. One is that docking is static and thus reflects only a singular pose of the interaction between the hCoV E peptide and the PALS1 protein at any given time. It does not account for any possible conformational changes that the E protein might undergo during its interaction with PALS1. Another is the length of the E peptides. Docking the full-length hCoV E protein to the full-length PALS1 protein would simulate a more accurate interaction as residues upstream of the PBM might contribute to host protein interaction [30,33,43]. After all, SARS-CoV-1 and 2 8-mer E peptides bind to PALS1 (29.6 µM and 22.8 µM, respectively) [40] with greater affinity than 12-mer peptides do (no interaction and 102 µM, respectively) [30], and the full-length SARS-CoV-1 E protein binds to syntenin, but the shorter 12-mer peptide does not [30]. Nevertheless, since our docking is validated and based on the experimentally determined PALS1 interactions with E peptides, we are confident in the data that we generated. We recommend the use of the full-length E proteins for future research. Future studies could also benefit from longer simulations to obtain more sampled conformations of the E protein structure in complex with human host partners. Furthermore, the experimental determination of the full-length structure of the E proteins will be a significant contribution to understanding human host protein interaction with hCoV E proteins. This study only provides a partial picture of the dynamics of the hCoV E proteins but shows good agreement between predictive modelling and simulation studies.

## 5. Conclusions

This is the first study to predict and generate full-length 3D models of the E protein for both the more virulent hCoVs (SARS-CoV-1, -2, and MERS-CoV) and the less virulent hCoVs (HCoV-229E and HCoV-NL63). This study also attempted to determine a distinguishing feature between the E proteins of the more virulent and the less virulent hCoVs to provide a possible explanation for the difference in pathogenicity and disease severity. Interaction partners have only been identified for the more virulent hCoV E proteins [30], and it remains to be seen if the less virulent hCoV E proteins can also interact with host cell proteins through their proposed PBMs, to what extent this could happen, and which host cell proteins they might interact with. Therefore, we docked 8-mer and 18-mer peptide C-terminal domains of five hCoV E proteins to the host PALS-1 protein to understand this protein-protein interaction. We observed that the SARS-CoV-2 PBM of the 8-mer and 18-mer peptides had more interactions with PALS1, possibly explaining its ability to associate with human host proteins resulting in a more severe pathogenicity. Admittedly, this would need to be verified experimentally through follow-up in vitro and in vivo studies to validate our in silico findings. The overall reduced dynamics displayed by the SARS-CoV-1, -2, and MERS-CoV E proteins with a flexible, extended coil at their PBMs could allow improved interaction with human host proteins to facilitate an increased viral pathogenicity [124]. Therefore, based on the key interacting residues formed between PALS1 and the E proteins of the more virulent hCoVs, and the absence of many of these interactions in the E proteins of the less virulent hCoVs, we hypothesize that the flexibility of the E protein PBM contributes to how it interacts with the PALS1 protein by virtue of the types of interactions, which in turn facilitates how stable the PALS1-E interaction is. In this way, the PBM can be considered a potential determinant of the virulence of the hCoV, distinguishing more virulent hCoVs from less virulent ones. This study attempted to illuminate the proposed mechanism of pathogenicity for a full-length protein structure including the PBM, albeit using a predictive model.

## Figures and Tables

**Figure 1 viruses-14-01707-f001:**
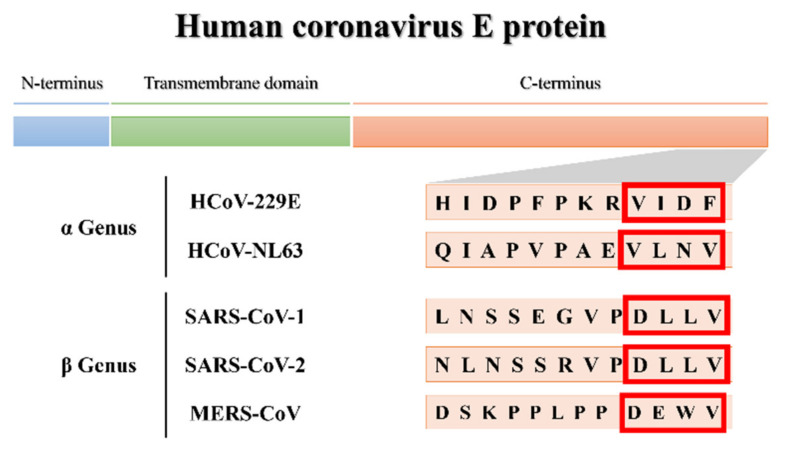
Partial C-terminus amino acid sequences of the human coronavirus (hCoV) envelope (E) proteins of the α- and β-CoV genera. Potential PDZ-binding motifs (PBMs) are indicated in the red blocks.

**Figure 2 viruses-14-01707-f002:**
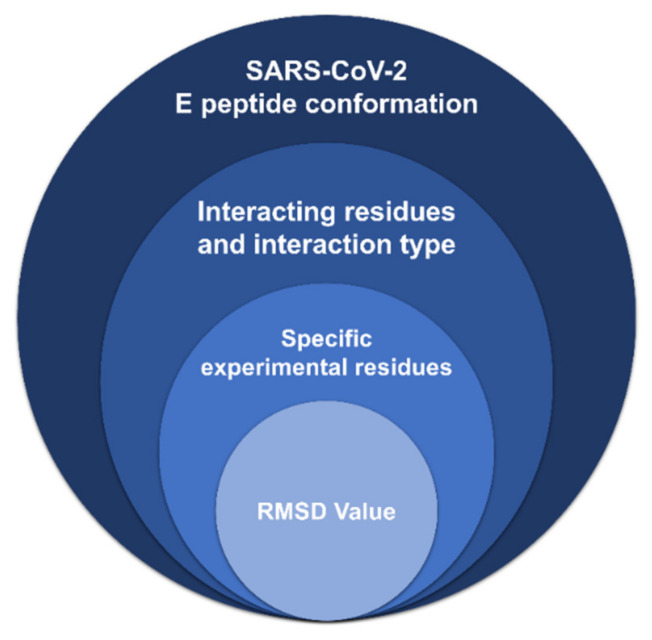
The selection criteria for the most accurately resembling docked protein cluster generated by the High Ambiguity Driven DOCKing (HADDOCK) webserver.

**Figure 3 viruses-14-01707-f003:**
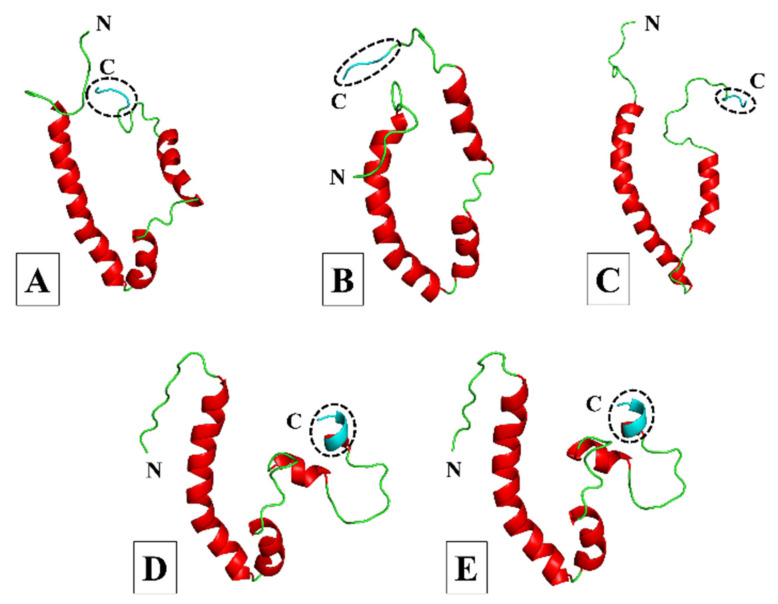
Cartoon representation of the three-dimensional (3D) models of the SARS-CoV-1, -2, MERS-CoV, HCoV-229E, and HCoV-NL63 envelope (E) proteins. (**A**) SARS-CoV-1. (**B**) SARS-CoV-2. (**C**) MERS-CoV. (**D**) HCoV-229E. (**E**) HCoV-NL63. Models were generated using MODELLER software and based on the partially resolved nuclear magnetic resonance (NMR) structures for SARS-CoV-1 E (PDB IDs: 5X29 and 2MM4) obtained from the Research Collaboratory for Structural Bioinformatics (RCSB) protein data bank (PDB) [49,50,51,52]. The models for SARS-CoV-1, -2 and HCoV-229E were generated from the 5X29 template, the MERS-CoV model was constructed from the 2MM4 template, and the HCoV-NL63 model was generated using the HCoV-229E E protein model as the template. The amino (N) and carboxy (C)-termini are indicated accordingly; coils are shown in green, α-helices in red, and the C-terminal PDZ-binding motif (PBM) in cyan and enclosed by a circle of dashed lines.

**Figure 4 viruses-14-01707-f004:**
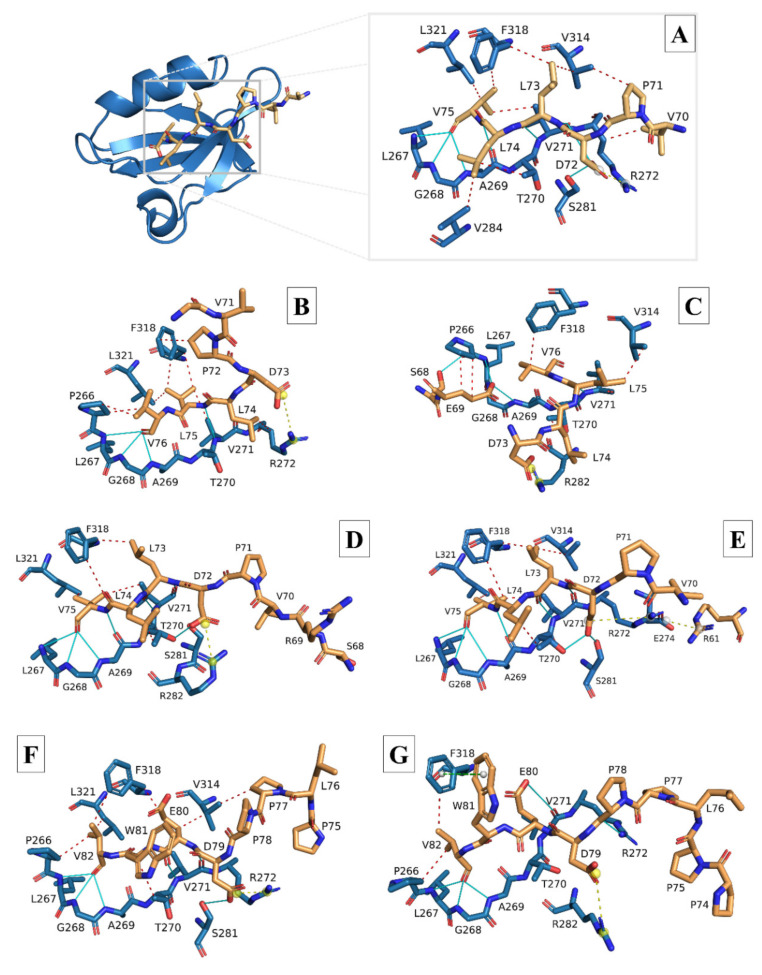
Stick models of the modelled 8-mer and 18-mer envelope (E) peptides of the more virulent hCoV E proteins docked to the PALS1 PDZ domain. (**A**) Ribbon and stick model of the SARS-CoV-2 experimental E peptide (stick, gold) docked to the PDZ domain of the PALS1 protein (ribbon and stick, blue) (PDB ID: 7NTK). (**B**) Stick model of the modelled SARS-CoV-1 (8-mer) E peptide (gold) docked to the PDZ domain of the PALS1 protein (blue). (**C**) Stick model of the modelled SARS-CoV-1 (18-mer) E peptide (gold) docked to the PDZ domain of the PALS1 protein (blue). (**D**) Stick model of the modelled SARS-CoV-2 (8-mer) E peptide (gold) docked to the PDZ domain of the PALS1 protein (blue). (**E**) Stick model of the modelled SARS-CoV-2 (18-mer) E peptide (gold) docked to the PDZ domain of the PALS1 protein (blue). (**F**) Stick model of the modelled MERS-CoV (8-mer) E peptide (gold) docked to the PDZ domain of the PALS1 protein (blue). (**G**) Stick model of the modelled MERS-CoV (18-mer) E peptide (gold) docked to the PDZ domain of the PALS1 protein (blue). Peptides in B-G were produced from the homology-modelled E proteins of the respective hCoVs and docked to the PDZ domain of the PALS1 protein (PDB ID: 7NTK) using the HADDOCK 2.4 webserver. Docked E peptides with the lowest RMSD values compared with the 7NTK experimental E peptide were uploaded to the protein–ligand interaction profiler (PLiP) webserver to demonstrate the types of interactions between PALS1 and the E peptides. Red, dashed lines: hydrophobic interactions; Cyan, solid lines: hydrogen bonds; Yellow, dashed lines: ionic interactions (salt bridges).

**Figure 5 viruses-14-01707-f005:**
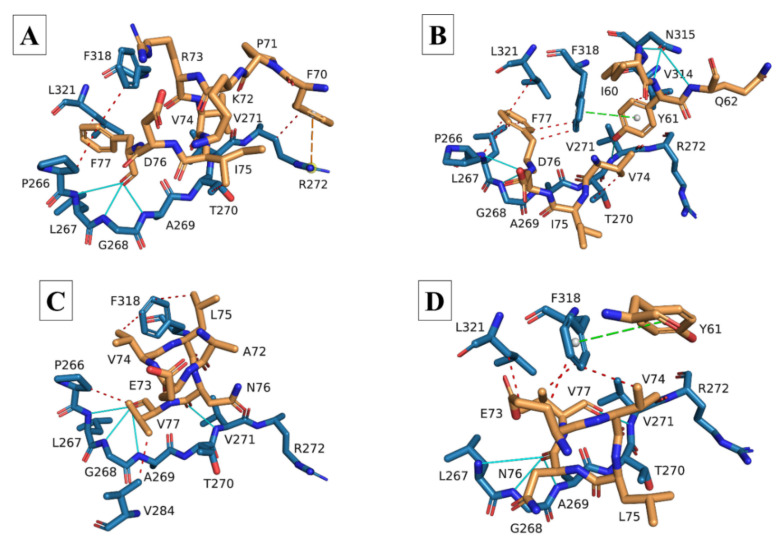
Stick models of the modelled 8-mer and 18-mer envelope (E) peptides of the less virulent HCoV-229E and HCoV-NL63 E proteins docked to the PALS1 PDZ domain. (**A**) Stick model of the modelled HCoV-229E (8-mer) E peptide (gold) docked to the PDZ domain of the PALS1 protein (blue). (**B**) Stick model of the modelled HCoV-229E (18-mer) E peptide (gold) docked to the PDZ domain of the PALS1 protein (blue). (**C**) Stick model of the modelled HCoV-NL63 (8-mer) E peptide (gold) docked to the PDZ domain of the PALS1 protein (blue). (**D**) Stick model of the modelled HCoV-NL63 (18-mer) E peptide (gold) docked to the PDZ domain of the PALS1 protein (blue). Peptides in (**A**–**D**) were produced from the homology-modelled E proteins of the respective hCoVs and docked to the partial PALS1 protein (PDB ID: 7NTK) using the HADDOCK webserver. Docked E peptides with the lowest RMSD values compared to the 7NTK experimental E peptide were uploaded to the protein–ligand interaction profiler (PLiP) webserver to demonstrate the types of interactions between PALS1 and the E peptides. Red, dashed lines: hydrophobic interactions; Cyan, solid lines: hydrogen bonds; Yellow, dashed lines: ionic interactions (salt bridges); Green, dashed lines: cation-π interactions.

**Figure 6 viruses-14-01707-f006:**
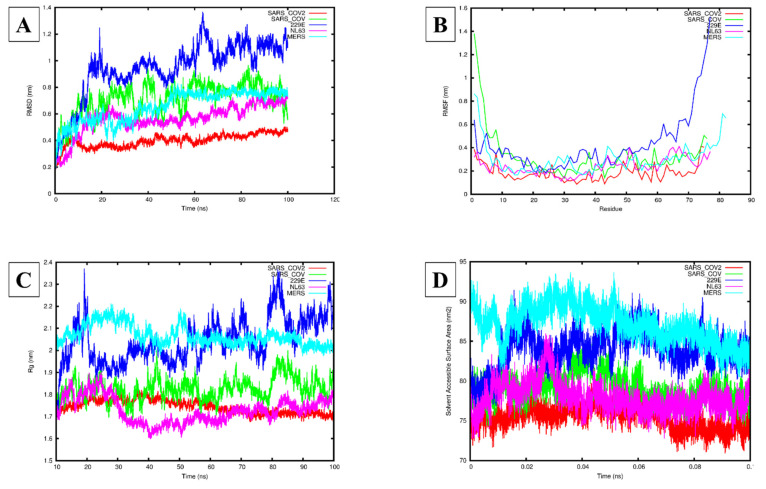
Molecular dynamics simulations of the human (h) coronavirus (CoV) envelope (E) proteins for SARS-CoV-1, -2, MERS-CoV, HCoV-229E, and HCoV-NL63. (**A**) The root mean square deviation (RMSD) (nm) of each of the modelled hCoV E proteins over 100 ns. (**B**) The root mean square fluctuation (RMSF) (nm) of each residue for the modelled hCoV E proteins. (**C**) The radius of gyration (Rg) (nm) of each of the modelled hCoV E proteins over 100 ns. (**D**) The solvent-accessible surface area (SASA) (nm^2^) of each of the modelled hCoV E proteins over 0.1 ns. Red: SARS-CoV-2 E protein, Green: SARS-CoV-1 E protein, Blue: HCoV-229E E protein, Pink: HCoV-NL63 E protein, Cyan: MERS-CoV E protein.

**Figure 7 viruses-14-01707-f007:**
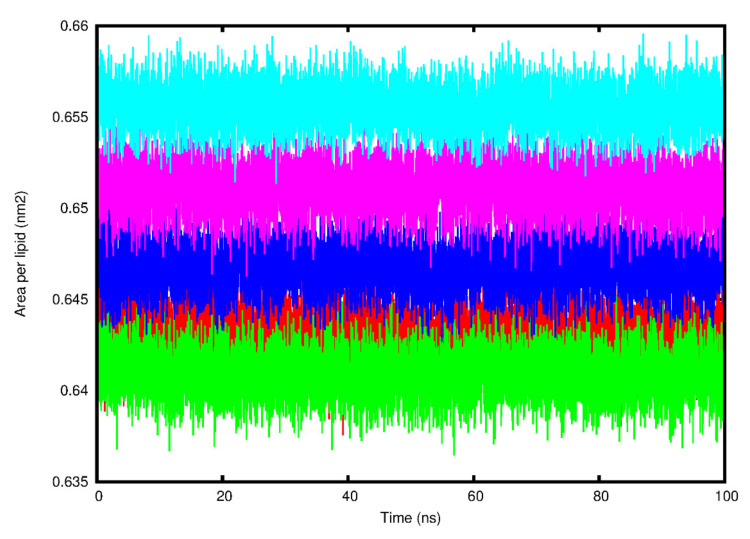
The area per lipid (nm^2^) fluctuation of the simulated 1-palmitoyl-2-oleoylphosphatidylcholine (POPC) lipid bilayer over the 100 ns time scale for the five human (h) coronavirus (CoV) envelope (E) protein systems. Red: SARS-CoV-2 E protein, Green: SARS-CoV-1 E protein, Blue: HCoV-229E E protein, Pink: HCoV-NL63 E protein, Cyan: MERS-CoV E protein.

**Table 1 viruses-14-01707-t001:** Types of interactions that occurred between the docked 8-mer hCoV E peptides and the PDZ domain of the PALS1 protein. The PALS1 PDZ domain residues to which the docked hCoV E peptides bound are shown at the top (second row) along with the corresponding PBM residue(s) of each docked hCoV E peptide. Residues are grouped according to the type of interaction that is formed between them.

HCoVs	Hydrophobic								Hydrogen						Ionic		π-Cation
	P266	T270	V271	R272	V284	V314	F318	L321	L267	G268	A269	T270	V271	S281	R272	R282	R272
**7NTK (8-mer)**	none	L74	V75	V70	L74	L73 P71	L73 V75	V75	V75	V75	V75 (×2)	none	L73 (×2) D72	D72	D72	none	none
**SARS-CoV-1**	V76 (×2)	none	L75	L74	none	none	L75 (×2) P72	L75	V76	V76	V76	none	L75	none	D73	none	none
**SARS-CoV-2**	none	L74	V75	none	L74	L73 (×2)	L73	V75	V75	V75	V75 (×2)	D72	L73 (×2)	D72	none	D72	none
**MERS-CoV**	V82	W81	none	none	none	P77 E80	E80 V82	V82	V82	V82	V82	none	E80 (×2)	D79 (×2)	D79	none	none
**HCoV-229E**	F77	I75	none	F70	none	none	F77	F77	F77	F77	F77	none	none	none	none	none	F70
**HCoV-NL63**	V77	none	none	none	V77	none	V74 L75	none	V77	V77	V77	none	N76	none	none	none	none

**Table 2 viruses-14-01707-t002:** Types of interactions that occurred between the docked 18-mer hCoV E peptides and the PDZ domain of the PALS1 protein. The PALS1 PDZ domain residues to which the docked hCoV E peptides bound are shown at the top (second row) along with the corresponding PBM residue(s) of each docked hCoV E peptide. Residues are grouped according to the type of interaction that forms between them.

HCoVs	Hydrophobic									Hydrogen									Ionic			π-Stacking
	P266	L267	T270	V271	R272	V284	V314	F318	L321	P266	L267	G268	A269	T270	V271	R272	S281	N315	R272	E274	R282	F318
**7NTK (8-mer)**	none	none	L74	V75	V70	L74	L73 P71	L73 V75	V75	none	V75	V75	V75 (×2)	none	L73 (×2) D72	none	D72	none	D72	none	none	none
**SARS-CoV-1**	E69 (×2)	none	L74	none	none	none	L75	V76	none	S68	E69	E69	E69	none	L75	none	none	none	none	none	D73	none
**SARS-CoV-2**	none	none	L74	V75	none	none	L73	L73 V75	V75	none	V75	V75	V75 (×2)	D72	L73 (×2)	none	D72	none	D72	R61	none	none
**MERS-CoV**	V82	none	none	none	none	none	none	V82	none	none	V82	V82	V82	none	E80	P78	none	none	none	none	none	W81
**HCoV-229E**	F77	F77 (×2)	V74	none	none	none	Y61 (×2)	F77 (×2)	F77	none	F77	F77	F77	none	Y61	none	none	I60 Y61 Q62	none	none	none	Y61
**HCoV-NL63**	none	none	none	none	none	none	none	E73 V74 V77	V77	none	N76	N76	N76	none	V77	none	none	none	none	none	none	Y61

**Table 3 viruses-14-01707-t003:** The lipid bilayer parameters of the 1-palmitoyl-2-oleoylphosphatidylcholine (POPC) simulated lipid bilayer and hydrogen bonds formed between the POPC lipids and the respective human (h) coronavirus (CoV) envelope (E) protein systems.

HCoV E Protein System	Area per Lipid (nm^2^)	Bilayer Thickness (nm)	Lateral Diffusion Coefficient (cm^2^/s)	Hydrogen Bonds
**SARS-CoV-1**	0.64 ± 0.01	4.39 ± 2.87	0.37 ± 0.02	10.58
**SARS-CoV-2**	0.64 ± 0.01	4.25 ± 2.48	0.43 ± 0.09	13.04
**MERS-CoV**	0.66 ± 0.01	5.78 ± 3.37	0.84 ± 0.01	9.22
**HCoV-229E**	0.65 ± 0.01	4.65 ± 2.71	0.46 ± 0.02	11.89
**HCoV-NL63**	0.65 ± 0.01	4.46 ± 2.60	0.34 ± 0.05	12.64

## Data Availability

Not applicable.

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
