# Peer review of "The Flexible, Extended Coil of the PDZ-Binding Motif of the Three Deadly Human Coronavirus E Proteins Plays a Role in Pathogenicity"

_viruses, 2022, doi:10.3390/v14081707_

Round 1
Reviewer 1 Report
Here Schoeman et al. are studying the Envelope protein from different coronaviruses from a structural point of view. The main results are the generation of in silico models for E proteins and the analysis of interaction with PALS1 protein (specifically PDZ domain).
I can understand the importance of studying this protein in different contexts, but here the results are weak, the manuscript needs to be completely rewritten, there is a lack of references, and the discussion needs to be revisited. The author's conclusions also are not fully supported by the results (e.g., "We observed that SARS-CoV-2 PBM 8-mer and 18-mer peptides had stronger affinity" - where are the binding affinity studies to support that? Is this only based on the number of interactions and conserved interactions (Tables 1 and 2)?). Since the authors are interested in C-terminal residues and those are not present in the templates used for homology modeling, why not use other approaches such as threading or ab initio? Or even AlphaFold?
Please, find below some additional comments:
- The English writing needs to be improved. Especially with respect to long sentences such as the following one:
Line 55-59: "The PBM of the prototypic SARS-CoV-1 E protein can bind to the PDZ domain of several host cell proteins interacts with the PDZ domain of the intercellular tight junction-associated protein protein-associated with lin-7 1 (PALS1) and redistributes PALS1 to the ER-Golgi intermediate compartment(ERGIC), where the E protein accumulates."
INTRODUCTION
- Introduction should have more references (e.g., lines 47-50 lines 87-89, ...)
- Line 93: change VS to vs
- In general, the introduction provides all the information necessary to understand the motivation of the authors in studying the envelope protein. However, it is not easy to read the text.
METHODS
- On the molecular modeling part, since 5X29 and 2MM4 are structures obtained from NMR, which conformation was used as a template?
- On lines 124-127, the references 38 and 39 refer to the tool, and not to the statement that "the structure of the missing C-terminal which were similar to the experimentally solved C-terminal E-peptide residues". Where is the comparison between the prediction and the experimental solved C-terminal E-peptide residues? The C-terminal residues are incomplete in the experimental structures (both 5X29 and 2MM4)?
RESULTS
- Results have repetitive text, which makes the reading unpleasant and hard to follow
- The homology modeling and quality assessment section could be resumed in one or two paragraphs
DISCUSSION
- Much of the discussion is a repetition of results and not actually a discussion
- There is a lack of reference for several sentences.
- In the phrase "And, bar the missing residues from 5x29 and 2mm4, all five of our E protein models share exactly the same structural features as those found in these previous studies, i.e. the N-terminus forms a coiled structure, the TMD adopts an a-helical conformation, and a second a-helix (H2) after the TMD [30,66,67], further validating the accuracy of our models." this does not validate the accuracy of your results, you still have the modeling bias. And these results are expected since you are using a homology modeling approach.
Author Response
I can understand the importance of studying this protein in different contexts, but here the results are weak, the manuscript needs to be completely rewritten, there is a lack of references, and the discussion needs to be revisited.
The author's conclusions also are not fully supported by the results (e.g., "We observed that SARS-CoV-2 PBM 8-mer and 18-mer peptides had stronger affinity" - where are the binding affinity studies to support that? Is this only based on the number of interactions and conserved interactions (Tables 1 and 2)?).
- Thank you kindly for pointing this oversight out. Please see lines 787-789 where we have clarified this to be based only on the number interactions showed in Tables 1 and 2.
Since the authors are interested in C-terminal residues and those are not present in the templates used for homology modeling, why not use other approaches such as threading or ab initio? Or even AlphaFold?
- We thank the reviewer for raising this valid point. We have compared the PBM region of our SARS-CoV-1, -2, and MERS-CoV E protein models to that of experimentally determined SARS-CoV-1 and -2 E protein PBMs such as those of PDB IDs 7NTJ, 7NTK, and 7M4R. The PBM region of these experimentally resolved structures adopted a similar flexible, coil conformation, leading us to conclude that our models adopted the appropriate conformation. We added Figure S3 in the supplementary material to show this – please see lines 256-263 where we refer to this.
Similarly, when we compared the modelled structure of our E protein PBM regions to an AlphaFold model and I-TASSER model, there was structural similarity in the secondary structure conformation. For the E protein models of HCoV-NL63 and HCoV-229E our template sequence did cover the PBM C-terminal motif.
Please, find below some additional comments:
- The English writing needs to be improved. Especially with respect to long sentences such as the following one:
Line 55-59: "The PBM of the prototypic SARS-CoV-1 E protein can bind to the PDZ domain of several host cell proteins interacts with the PDZ domain of the intercellular tight junction-associated protein protein-associated with lin-7 1 (PALS1) and redistributes PALS1 to the ER-Golgi intermediate compartment (ERGIC), where the E protein accumulates."
- We thank the reviewer for pointing this out. We apologise for this error and have amended it to read better – please see lines 57-63.
INTRODUCTION
- Introduction should have more references (e.g., lines 47-50 lines 87-89, ...)
- Thank you for this suggestion. Please see the added references to lines 48-52 and 81-83.
- Line 93: change VS to vs
- Changed at line 96.
- In general, the introduction provides all the information necessary to understand the motivation of the authors in studying the envelope protein. However, it is not easy to read the text.
- We do apologise for this and thank the reviewer for taking the time to point this out. We have amended the introduction to improve readability.
METHODS
- On the molecular modeling part, since 5X29 and 2MM4 are structures obtained from NMR, which conformation was used as a template?
- Please see lines 122-127 where we explain which PDB IDs were used as templates for the generation of our E protein models.
Briefly, 5x29 was used as a template to construct models for the SARS-CoV-1, -2, and HCoV-229E E proteins, 2mm4 was used to construct the E protein model for MERS-CoV.
Since the HCoV-NL63 E protein has a much higher sequence identity with the HCoV-229E E protein, the E protein model of HCoV-229E was used to generate the model for the HCoV-NL63 E protein. The templates used were selected based on the highest sequence identity obtained from a multiple sequence alignment, which we show in the supplementary material as Figure S1.
- On lines 124-127, the references 38 and 39 refer to the tool, and not to the statement that "the structure of the missing C-terminal which were similar to the experimentally solved C-terminal E-peptide residues". Where is the comparison between the prediction and the experimental solved C-terminal E-peptide residues? The C-terminal residues are incomplete in the experimental structures (both 5X29 and 2MM4)?
- We thank the reviewer for drawing our attention to this. We originally used the references at the end of that sentence to reference MODELLER but we have amended the sentence to avoid confusion by placing the references earlier – please see lines 129-132. We have also added a figure to the supplementary material (figure S3) to show the structural superimposition between our missing modelled C-terminal residues and the experimentally determined SARS-CoV-1 and -2 E protein C-terminal residues. Please also see figure S1 in the supplementary materials that shows the multiple sequence alignment between each template and the corresponding E protein model that was produced from it.
RESULTS
- Results have repetitive text, which makes the reading unpleasant and hard to follow
- Homology modelling and quality assessment section could be resumed in one or two paragraphs
- We thank the reviewer for pointing this out. We have made some adjustments to the Results section to avoid repetition and improve readability. We do, however, prefer to keep the results between the more virulent hCoVs and the less virulent ones separate as we aim to highlight the distinction(s) between them.
DISCUSSION
- Much of the discussion is a repetition of results and not actually a discussion.
- There is a lack of reference for several sentences.
- We appreciate the reviewer pointing this out, thank you. We have made some adjustments to avoid a repetition of the results and included some additional references where possible. We would like to bring to the reviewer’s attention that the human coronavirus E protein has not been researched as well as the other coronaviral structural proteins, which is why we are not always able to refer to many other studies. As the first hCoV to cause a severe outbreak, SARS-CoV-1 received the most attention; less attention was given to the MERS-CoV outbreak since it was more confined; with SARS-CoV-2 being the first human coronavirus to cause a pandemic, the interest in coronaviruses was renewed and the E protein is now getting much more attention.
We referenced studies that were the most relevant to our findings and managed to include some more as suggested.
- In the phrase "And, bar the missing residues from 5x29 and 2mm4, all five of our E protein models share exactly the same structural features as those found in these previous studies, i.e. the N-terminus forms a coiled structure, the TMD adopts an a-helical conformation, and a second a-helix (H2) after the TMD [30,66,67], further validating the accuracy of our models." this does not validate the accuracy of your results, you still have the modelling bias. And these results are expected since you are using a homology modelling approach.
- We thank the reviewer kindly for pointing this out, and we apologise for this oversight. We have amended this at lines 542-547 to reflect more accurately that our findings compare with previous predictions based on the secondary structure arrangement of our models.
Reviewer 2 Report
Manuscript Title: The flexible, extended alpha helix coil of the PDZ-binding motif of the three deadly human coronavirus E proteins plays a role in pathogenicity
Major comments
1. Authors have mentioned that the PBM-containing C-terminal peptides of SARS-CoV-2 E showed that SARS-CoV-2 E can 69 interact with PALS1 in a similar way to what SARS-CoV-1 E does. In which mechanism SARS-CoV-2 shows more virulence than SARS-CoV-1, if authors explain more in detail it could be more informative.
2. Authors have mentioned that about 5 stains were less virulent but for the current study thay have selected only three stains to CoVs 229E and NL63 E; is there any specific reason for selecting particularly these three strains?
3. In the molecular modeling section; SARS-CoV-1 E protein (PDBIDs: 5x29) was used to generate full-length 3D models of the E proteins for SARS-CoV-1, -2 and structures lacking some of the C-terminus.
Some papers have published with full models with multiple chains, kindly check
https://www.sciencedirect.com/science/article/pii/S1319562X21002096
if that is the case, authors can try using the multi-chain models.
4. Did authors have done any other validation analysis to check the overall quality of the predicted model along with PROCHECK?
5. The detailed abbreviations of some short forms are not provided in the manuscript and it makes reading the manuscript difficult.
6. In figure 3 authors have marked the protein with a dotted line, but no information on what it does represents in the structure. Provide this information more in detail.
7. Why is the POPC membrane used here, any reasons? If so, kindly quote those relations with this target enzyme.
8. How the authors positioned ions in the POPC and water model, how charges are validated in this, any idea?
9. How those interactions of membranes and enzymes are related, it will be interesting to show those in results and discussion, as the authors have provided only the RMSD and RMSF data is not satisfactory, especially in those short timescale events (In presence of POPC membrane).
10. I suggest authors replace that 2D interaction pattern with 3D figures with good clarity or at least 600 DPI.
Author Response
Major comments
- Authors have mentioned that the PBM-containing C-terminal peptides of SARS-CoV-2 E showed that SARS-CoV-2 E can 69 interact with PALS1 in a similar way to what SARS-CoV-1 E does. In which mechanism SARS-CoV-2 shows more virulence than SARS-CoV-1, if authors explain more in detail it could be more informative.
- We thank the reviewer for bringing up this point. In lines 657-660 we point out that the SARS-CoV-2 E peptide forms more interactions with PALS1 than the SARS-CoV-1 E peptide does, suggesting that the interaction between the SARS-CoV-2 E protein and PALS1 might be more stable than the interaction between the SARS-CoV-1 E protein and PALS1. We also added the reference (lines 660-667) to a study (Lo Cascio, et al., reference [41], DOI: https://doi.org/10.1016/j.csbj.2021.03.014) that performed 4 runs of molecular dynamics simulations with the SARS-CoV-1 E peptide bound to the PALS1 protein. They reported that the SARS-CoV-1 E peptide detached from the PALS1 protein in all 4 runs of molecular dynamics simulations. We also highlight the possibility in lines 761-766 that residues upstream of the PDZ-binding motif might contribute to the binding of the E protein to host proteins. And since there are 4 mutation differences between the SARS-CoV-1 and -2 E proteins, these differences might also allow SARS-CoV-2 to form host protein interactions that can make it more severe than SARS-CoV-1.
- Authors have mentioned that about 5 stains were less virulent but for the current study they have selected only three stains to CoVs 229E and NL63 E; is there any specific reason for selecting particularly these three strains?
- We thank the reviewer for this question. Please refer to lines 529-532 where we explain that we attempted to generate 3D models for the E proteins of both HCoV-OC43 and HCoV-HKU1 but that the sequence identity between the templates and these models was too low. This would have generated models for OC43 and HKU1 that had very low accuracy and would not have been reliable. We, therefore, only generated 3D models for the HCoV-229E and HCoV-NL63 E proteins. We could employ ab initio methods in future studies to construct more reliable models for the E proteins of OC43 and HKU1.
- In the molecular modeling section; SARS-CoV-1 E protein (PDBIDs: 5x29) was used to generate full-length 3D models of the E proteins for SARS-CoV-1, -2 and structures lacking some of the C-terminus.
Some papers have published with full models with multiple chains, kindly check
https://www.sciencedirect.com/science/article/pii/S1319562X21002096
if that is the case, authors can try using the multi-chain models.
- We appreciate that the reviewer pointed us to this paper. However, this paper generated a model for SARS-CoV-2 E protein using only the transmembrane domain (TMD) from PDB ID: 7K3G and some additional residues from PDB ID: 2MM4 as templates. Since both 7K3G and 2MM4 also lacked the C-terminal residues, we did not consider this paper as having published a full-length model. They did, however, use MODELLER to construct their model as per the current study.
- Did authors have done any other validation analysis to check the overall quality of the predicted model along with PROCHECK?
- We thank the reviewer for enquiring about this. We have included additional validation analysis in the form of ProSA Z-scores and GA341 scores for the models:
- Description of scores (Methods): lines 138-143
- Scores (Results): lines 271-275, 290-293, and 514-518
- The detailed abbreviations of some short forms are not provided in the manuscript and it makes reading the manuscript difficult.
- We thank the reviewer for taking the time to bring this to our attention and apologise for this oversight. We have ensured that all short forms have been abbreviated in detail before being carried through the rest of the manuscript.
- In figure 3 authors have marked the protein with a dotted line, but no information on what it does represents in the structure. Provide this information more in detail.
- We appreciate the reviewer pointing this out. We apologise for this oversight and have added a description of the dotted line in the caption of figure 3 at line 304.
- Why is the POPC membrane used here, any reasons? If so, kindly quote those relations with this target enzyme.
- We thank the reviewer for raising this question. The coronavirus envelope protein is a membrane protein with one transmembrane domain (TMD) that inserts into the membrane of infected cells. Therefore, MD simulations were performed in a POPC membrane to obtain a more realistic simulation of the protein. A POPC membrane was used specifically since POPC is the most abundant lipid in mammals and eukaryotic membranes. We have included this information in the revised manuscript at lines 204-207.
- How the authors positioned ions in the POPC and water model, how charges are validated in this, any idea?
- The CHARMM-GUI web interface was used to prepare each of the E protein models for the MD simulations, and the Monte-Carlo ion placing method was selected during the preparation process. This method randomises the ion placement in the water box that is used to solvate the protein. Please see lines 200-212 where we explained and clarified this.
- How those interactions of membranes and enzymes are related, it will be interesting to show those in results and discussion, as the authors have provided only the RMSD and RMSF data is not satisfactory, especially in those short timescale events (In presence of POPC membrane).
- We appreciate the reviewer highlighting this point. Here, we performed POPC lipid bilayer analysis, looking at the area per lipid, bilayer thickness, lateral diffusion and deuterium order parameters. Please see lines 237-243 for this. To understand protein-membrane interaction, we also calculated the number of hydrogen bonds between the protein and the POPC lipid bilayer as well as polar contacts formed between the PBM motif and the POPC lipid bilayer. Please see lines 243-247 for this. The results for this analysis are reported in lines 447-467 and the discussion of the interpretation of the results are reported in lines 591-595.
- I suggest authors replace that 2D interaction pattern with 3D figures with good clarity or at least 600 DPI.
- We thank the reviewer for bringing this to our attention and apologise for this oversight. We generated higher-quality images for the 2D interaction figures and uploaded them with all other figures at the highest quality we could.
Reviewer 3 Report
In the manuscript of Schoeman et al. entitled “The flexible, extended alpha helix coil of the PDZ-binding motif of the three deadly human coronavirus E proteins plays a role in pathogenicity” the authors generated full-length E protein models for SARS-CoV-1, SARS-CoV-2, MERS-CoV, HCoV-229E, and HCoV-NL63 and docked C-terminal peptides of each model to the PDZ domain of the human polarity protein associated with lin seven 1 (PALS1). The PDZ-binding motif (PBM) of the SARS-CoV-1, -2, and MERS-CoV models adopted a more flexible, extended coil while the HCoV-229E and HCoV-NL63 models adopted a less flexible alpha helix. All the E peptides docked to PALS1 occupied the same binding site and the more virulent hCoV E peptides generally interacted more stably with PALS1 than the less virulent ones. Thus, the authors propose that the increased flexibility of the PBM in more virulent hCoVs may permit more stable binding to various host proteins, thereby possibly contributing to more severe disease. The authors concluded that this is the first paper to model full-length 3D structures for both more virulent and less virulent hCoVs E proteins, providing novel insights for possible drug and/or vaccine development. This study is interesting and should be accepted after minor revisions.
Minor revisions
11) There are many other limitations that the study does not explore: (i) the number of genomes sequenced and available among the viruses studied is enormous; (ii) the authors need to indicate a practical application for this in-silico analysis. For example, discuss and make available a tool that can analyze the gene encoding the E protein to verify if new SARS-CoV-2 variants are genetically evolving to become less pathogenic? (iii) to evaluate these changes in vitro and in vivo to verify if the in-silico analyses make sense.
22) The authors need to describe the difference between this study and other published studies:
2.1) https://doi.org/10.1371/journal.pone.0251955
2.2) https://doi.org/10.1038/s42003-021-02250-7
2.3) https://doi.org/10.2217/fvl-2020-0365
2.4) https://doi.org/10.1371/journal.ppat.1004320
Author Response
The authors concluded that this is the first paper to model full-length 3D structures for both more virulent and less virulent hCoVs E proteins, providing novel insights for possible drug and/or vaccine development. This study is interesting and should be accepted after minor revisions.
Minor revisions
- There are many other limitations that the study does not explore: (i) the number of genomes sequenced and available among the viruses studied is enormous;
- We appreciate the reviewer raising this valid point. We have addressed this issue and included references to papers that report minimal to no mutations in the E gene of SARS-CoV-2 sequenced genomes – please see lines 519-527.
- (ii) the authors need to indicate a practical application for this in-silico analysis.
For example, discuss and make available a tool that can analyze the gene encoding the E protein to verify if new SARS-CoV-2 variants are genetically evolving to become less pathogenic?
- We thank the reviewer for bringing this to our attention. We have included potential applications for our E protein models in lines 751-756.
- (iii) to evaluate these changes in vitro and in vivo to verify if the in-silico analyses make sense.
- We thank the reviewer for raising this point. However, we feel this is something that can be addressed in follow-up studies and even looking at sequencing data. But this is beyond the scope of this current study.
- The authors need to describe the difference between this study and other published studies:
- 1) https://doi.org/10.1371/journal.pone.0251955
- 2) https://doi.org/10.1038/s42003-021-02250-7
- 3) https://doi.org/10.2217/fvl-2020-0365
- 4) https://doi.org/10.1371/journal.ppat.1004320
- We appreciate the reviewer bringing these studies to our attention and we identified 3 major differences between these studies and our study. These differences are addressed at lines 729-751.
Briefly, (1) while other papers constructed only partial models of the E protein, our paper constructed full-length models.
(2) Our paper compared the ability of the more virulent and less virulent hCoV E proteins to bind to the PALS1 host protein, whereas the suggested papers only looked at SARS-CoV-1 E or compared SARS-CoV-1 E with SARS-CoV-2 E binding to the PALS1 protein.
(3) Our paper constructed models of both the more virulent and the less virulent hCoV E proteins, whereas other papers only modelled the more virulent SARS-CoV-1 and -2 E proteins, and (one paper) the MERS-CoV E protein. Our study generated a full-length model of the MERS-CoV E protein as well as a model for the E proteins of the less virulent HCoV-229E and HCoV-NL63. This is something that has not been done for the latter 2 hCoVs.
Round 2
Reviewer 1 Report
The authors improved the paper and it could be accepted after minor revisions.
Reviewer 2 Report
Revised version can be accepted for publication
Best wishes for the authors